# Ecological rules for global species distribution also predict performance variation in Ironman triathletes

**Ryan Calsbeek**◉*

Department of Biological Sciences, Dartmouth College, Hanover, NH, United States of America

* Ryan.calsbeek@dartmouth.edu

**Data Availability Statement:** The data file is available from the Dryad data base (accession: 10.5061/dryad.573n5tbbw).

## Abstract

Bergmann's and Allen's rules predict changes in body size and appendage length across temperature gradients for species with broad geographic distributions. Larger bodies and longer limbs facilitate cooling whereas smaller bodies and compact limbs limit heat loss. Although these patterns are highly repeatable (hence "rules" of ecology) the patterns and underlying mechanisms are less-well understood in humans. Here I show that variation in running performance among human male triathletes is consistent with both Bergmann's and Allen's rules. Males (but not females) with relatively larger body size and longer limbs performed better at hot compared to cold race venues and vice-versa. Consistent with results in other taxa, sex-specificity may reflect selection for sexual dimorphism. Results suggest that ecological patterns detected over large-spatial scales may arise from fine-scale variation in locomotor performance.

## Introduction

Ecology's power to explain patterns across multiple spatial scales has produced fundamental insights into the principles that structure biological communities. These include grand scale descriptions of the processes that maintain biological diversity (e.g., frequency dependent selection within and among species [1]) as well as simple observations regarding patterns of species distribution around the globe [2]. Although abundant empirical evidence illustrates the generality of ecological rules [3–5], debate persists regarding the causal mechanisms that operate across different scales to create these patterns.

Bergmann's and Allen's rules are two powerful examples of ecological pattern that manifest across multiple spatial scales. These 'rules' describe relationships between body size (Bergmann) and shape (Allen) that vary across temperature gradients. In general, taxa with broad geographical distributions tend to exhibit compact bodies in cooler environments and long lanky bodies in hotter environments [6, 7]. Taxa in cooler environments tend to be likewise characterized by shorter appendages compared to their relatively long-limbed counterparts in hotter environments [8]. Bergmann's original (1847) hypothesis suggested that variation in body size arose due to changes in the ratio of surface area to volume that accompany differences in body plan. More compact body forms have reduced surface area to volume ratios and

**Funding:** The authors received no specific funding for this work.

are better suited to cool climates because of their superior heat retention. Elongate body plans should be adaptive in warmer climates where they are better able to dissipate heat [9]. Despite 180 years of study and repeated demonstration of the patterns behind Bergmann's and Allen's rules, tests of the thermoregulatory hypothesis remain surprisingly rare [10].

Similar to data from other taxa, human populations, both ancient and extant, exhibit variation in body size and limb proportions consistent with both Bergmann's and Allen's rules [11, 12]. Whether the same causal mechanisms underlie human variation in body plan remains an open question. Alternative explanations range from northern hemisphere bias in sampling [13], to morphological plasticity arising from diet and pathogen load [14]. Nevertheless, biogeographic patterns in human morphology are consistent with the influence of natural selection arising from variation in locomotor performance, particularly endurance running performance, in different environments [15, 16]. Moreover, although humans have long been considered poor runners compared to other mammals [17], there is growing appreciation for the fact that our capacity for endurance running matches that of many other mammalian runners, and far exceeds that of any other primate [16].

Several anatomical differences between humans and other primates are consistent with natural selection favoring improved endurance running, especially running in hot climates [18]. An elongate body plan, with narrow pelvic girdle, broad shoulders, and long legs serve to facilitate greater stride length and decouple the rotation of the upper body from the hips [19]. Changes in the distribution of connective tissues reduce costs of transport by storing and releasing elastic energy during locomotion. The distribution of adipose tissue and sweat glands in humans enhances thermoregulatory capacity [20]. These locomotor adaptations may be sexually dimorphic since human females, on average, have smaller body plans with greater surface area, more sweat glands and more adipose tissue [21]. Taken together these observations suggest that humans have experienced natural selection for improved endurance running performance compared to other primates, and that this selection may have differed between males and females. If true, then the sex-specificity of locomotor function may have as yet unexplored consequences for Bergmann's and Allen's rules in humans.

Here, I investigate whether variation in human athletic performance is consistent with the predictions of Bergmann's and Allen's rules. I test how performance in professional triathletes varies as a function of body morphology and ambient temperature. Previous studies linking human performance variation, morphology, and temperature are rare but suggestive. In one study, athletes with relatively longer legs were more likely to *finish* endurance running events in hot conditions whereas athletes with shorter legs were more likely to finish similar events in cool conditions [22]. To date however, no study has reported quantitative variation in both performance and morphology for male and female athletes competing in different environmental conditions as a test of Bergmann's and Allen's rules in humans. Here I take new steps towards addressing that shortcoming.

Triathlons are athletic competitions in which individual athletes compete in consecutive swimming, bicycling, and running events. Among the longest of these events is the so-called Ironman™ triathlon in which participants swim 3.8 Km, cycle 180 Km, and run 42.2 Km. Iron-distance triathlons are held at venues around the world providing an opportunity to compare athletic performance of the same athletes in multiple thermal contexts. At least two aspects of triathlon are especially amenable to the study of human athletic performance. First, professional triathlons provide a novel solution to sample size limitations that normally arise from athlete-reluctance to volunteer for measurement on the day of competition. Triathlon event photographers publish numerous photographs of professional triathletes from the cycling portion of the event on the internet, and although athletes ride bicycles of various sizes, wheel diameters are identical (622mm), providing a convenient size standard for measurement *in*

*silico*. Second, swimming and cycling performance measures, in contrast to running performance, serve as a pair of null model tests for correlations between morphology, performance, and temperature since neither activity would have been important in the evolution of early humans.

I acquired performance data from the publicly available Ironman™ triathlon database at www.ironman.com and from https://protriathletes.org/. I included only athletes that had competed in at least two Ironman distance triathlons and for whom I could find high resolution images from the cycling portion of the event taken from an angle orthogonal to the direction of travel. This allowed me to identify consistent landmarks on the body from which morphological measures could be recorded. I measured lengths of the torso, upper and lower leg, upper and lower arm (Fig 1) using the measurement tools in ImageJ software v. 1.53. Data on height were available for 90% of male and 78% of female athletes from the professional triathlete's organization (PTO, stats.protriathletes.org/). Missing values for height were estimated from the from the regression of these known body heights on total leg length ($r^2 = 0.43$). I corroborated these estimated values against measured estimates from digital images. I opted to use the regression-based estimates for analysis because cycling helmets obscured the ability to locate the top of the head and the length of the neck. As a conservative approach I also performed all analyses after excluding these estimated values. In all cases, results were qualitatively

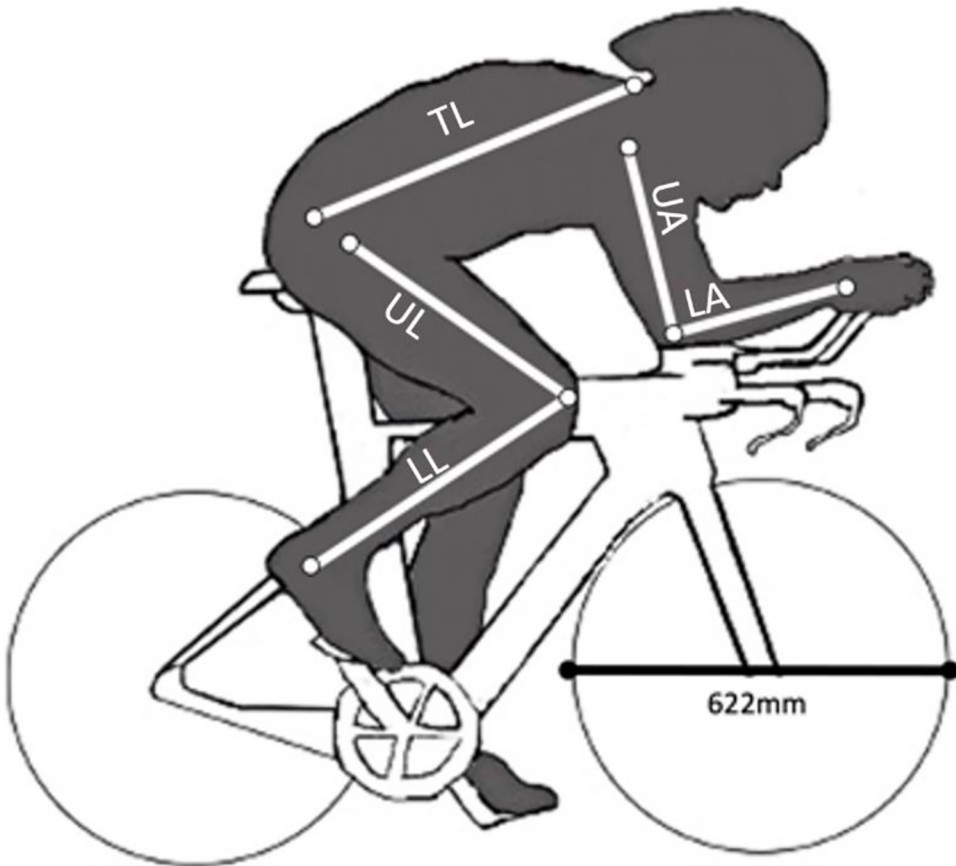

**Fig 1. Representative silhouette of a triathlete measured in this study.** Road-bicycle wheel diameter (622mm) was used as a size standard in the digital image analysis software ImageJ. The traits measured are indicated by white lines with approximate landmarks shown with open circles. Abbreviations are as follows: TL = torso length, UA = upper arm, LA = lower arm, UL = upper leg, LL = lower length. The sums of upper and lower arm and leg lengths were used in analyses.

identical, and all significant results remained statistically significant with the smaller data set. The complete data set yielded a sample of 173 total athletes (98 males and 75 females).

Because limb length is correlated with body size, simultaneously testing the predictions of Allen's and Bergmann's rules requires decoupling this relationship. I therefore estimated relative arm and leg lengths by regressing raw values on total body height and saving the residuals from these separate regressions. These values represent mathematically independent measures of limb length relative to body size for use in evaluating Bergmann's rule. Conceptually, body shape is a more obtuse measure of shape. Several alternative indices of body shape have been proposed (e.g., body mass index, ponderal index), none of which address the multicollinearity of body size and limb length. To account for this problem, I created two principal component scores using torso length, upper and lower arm lengths, upper and lower leg lengths, and height. All measures weighed positively on PC1 which can therefore be considered a general description of body size. By contrast, whereas body mass and height had positive weights on PC2, the weighting of arm and leg lengths were negative. PC2 can therefore be considered a measure of body shape [for use in evaluating Allen's rule] that is mathematically independent of the size axis captured by PC1.

I recorded two measures of locomotor performance at each event. First, I recorded the time (seconds) to complete each of the three events as well as the total race time (total race time is the sum of the three individual events, plus the time needed to transition between events). Second, to account for variation in performance due to aspects of the racecourse itself (e.g., course topography, altitude, etc..). I recorded relative performance for male and female athletes by dividing their individual race times by the average race times of the other professional male and female triathletes respectively at each race venue (no age-group/amateur athletes were included in this study owing to the lack of consistently available photographs). Finally, I recorded the daily high temperature for each race venue on the day of competition using historical temperature records available at www.wunderground.com/.

To test the hypothesis that variation in body shape leads to temperature-dependent performance variation that is consistent with Bergmann's rule, I used Repeated Measures Analyses of Variance in which the dependent variables were run times measured at each of two Iron-distance triathlon events and the independent variables were PC2 and the difference in temperature between the two event venues (hereafter "ΔT") as well as their interaction. To test the hypothesis that variation in appendage length leads to temperature-dependent performance variation that is consistent with Allen's rule, I again used Repeated Measures Analyses of Variance in which the dependent variables were running performance measured at each of two Iron-distance triathlon events, and the independent variables were PC1, difference in temperature between the two event venues ("ΔT") and the interaction between PC1 and delta T. I further tested for the individual effects of residual arm and leg lengths in place of PC1 to illustrate the contribution of each appendage measure to performance. I initially tested for sex specificity in the data. In tests of both Allen's and Bergmann's rule, I found significant sex × temperature × performance interactions. As such, all subsequent analyses were conducted separately for males and females. Where statistically significant, I visualized these interaction terms using three-dimensional surface fitting. All analyses were performed using JMP Pro v. 16.0.0.

## Results

Values for morphological traits measured in this study were all larger for males than for females (Table 1; all $P < 0.0002$). The average total time ($\bar{X} \pm SE$) to event completion for females (9hrs. 31 minutes = 34237 seconds (± 161)) was significantly longer than that of males (8hrs. 44 minutes = 31498 seconds (± 123)) (ANOVA $F_{1,170} = 188.11$, $P < 0.0001$). Likewise

**Table 1. Morphological trait values measured in this study.**

|                  | female range | mean (SE)   | male range  | mean (SE)   |
|------------------|--------------|-------------|-------------|-------------|
| Height (m)       | 1.52–1.78    | 1.69 (.006) | 1.65–1.94   | 1.81 (0.006)|
| Mass (kg)        | 50–68        | 57.2 (.61)  | 62–85       | 73 (0.45)   |
| upper leg (mm)   | 342–517      | 435 (4.02)  | 382–607     | 484 (4.27)  |
| lower leg (mm)   | 341–491      | 404 (3.54)  | 386–549     | 446 (3.32)  |
| upper arm (mm)   | 244–368      | 300 (2.81)  | 265–482     | 337 (3.25)  |
| lower arm (mm)   | 195–314      | 240 (2.79)  | 211–365     | 274 (2.59)  |
| Torso (mm)       | 458–592      | 525 (3.50)  | 492–696     | 591 (4.49)  |

finish times for the three individual events were longer for females compared to males (swim: 3520 s. (± 24.9) versus 3168 s. (± 21.9), ANOVA $F_{1,170}$ = 112.27, P<0.0001; bike: 18307 s. (± 96.19) versus 16581 s. (± 84.59), ANOVA $F_{1,170}$ = 181.59, P<0.0001; run: 12070 s. (± 63.61) versus 11425 s. (± 55.93), ANOVA $F_{1,170}$ = 57.98, P<0.0001). For both sexes, total race times are slightly longer than the sum of individual event times owing to the transition times required between events. Transition times were not included in any analyses. Finally, the ambient temperature for races analyzed in this study ranged from 17.78°C to 38.89°C (mean = 27.37± 0.32) and ΔT ranged from 0°-21.1° C (mean 6.43 ± 21.11).

## Allen's rule

Changes in running performance were consistent with the predictions from Allen's rule in males but not females. Total change in relative running performance between venues varied with our overall measure of size (PC1) and race day temperature in males (rmANOVA PC1 × ΔT: $F_{1,93}$ = 9.65, P = 0.003 Fig 2A), but not females (rmANOVA PC1 × ΔT: $F_{1,71}$ = 1.11, P = 0.27; Fig 2C). This difference was statistically significant (PC1×ΔT×sex, P<0.01). Male athletes with longer arms and legs (both relative to body size and absolute length) ran faster in hotter conditions, whereas males with shorter arms and legs ran faster in cooler conditions. Individual morphological metrics were significant for leg length and temperature (rmANOVA on relative leg length × ΔT: $F_{1,93}$ = 7.06, P = 0.009 and for absolute leg length × ΔT: $F_{1,93}$ = 8.63, P = 0.004) as well as for arm length (rmANOVA on relative arm length×ΔT: $F_{1,93}$ = 6.24, P = 0.01 and absolute arm length × ΔT $F_{1,93}$ = 9.13, P = 0.003). None of these results were significant in female athletes, although results were qualitatively similar for female leg length (rmANOVA on relative leg length × ΔT: $F_{1,71}$ = 2.18, P = 0.14 and for absolute leg length × ΔT: $F_{1,71}$ = 3.17, P = 0.08) (all other P>0.27). Changes in swimming and cycling performance as a function of size and temperature were not significant in either sex (Table 2).

## Bergmann's rule

Changes in running performance were likewise consistent with the predictions from Bergmann's rule in males but not females. Total change in relative running performance between venues varied with our overall measure of body shape (PC2) and race day temperature in males (rmANOVA PC2 × ΔT: $F_{1,93}$ = 4.72, P = 0.03; Fig 2B) but not in females (rmANOVA PC2 × ΔT: $F_{1,71}$ = 1.36, P = 0.25; Fig 2D) and this difference was likewise significant (PC2×ΔT×sex, P<0.03). Changes in swimming and cycling performance as a function of body shape and temperature were not significant in either sex (Table 2).

## Discussion

After more than 180 years of study and multiple demonstrations of their respective biogeographic predictions, Bergmann's and Allen's rules remain controversial [4, 10–14]. This

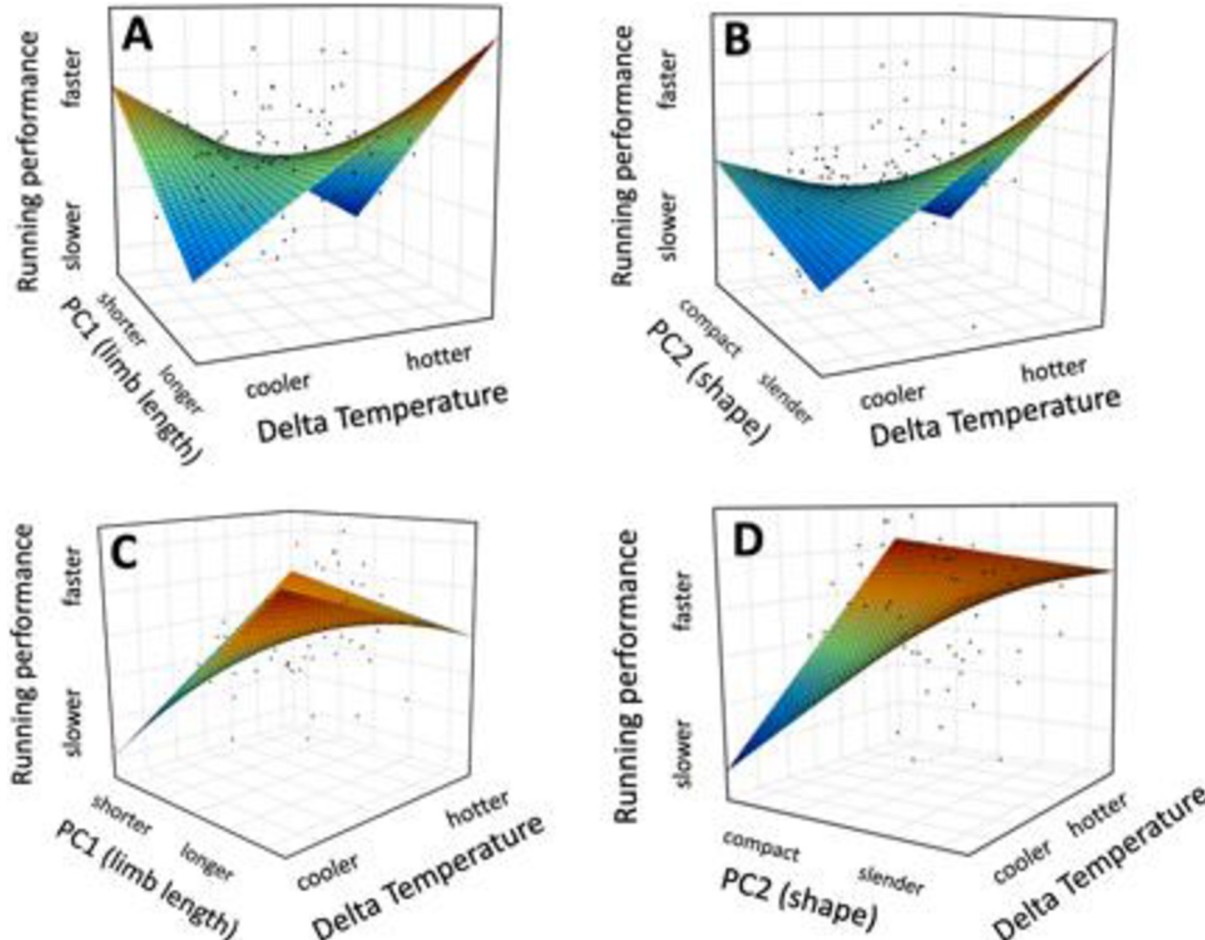

**Fig 2.** Parametric performance surfaces for males (top row) and females (bottom row). Male running performance varied significantly as a function of the interaction between morphology and temperature. Male triathletes with longer arm and leg lengths (relative to body height) ran faster at hotter race venues, whereas males with shorter relative arm and leg lengths ran faster at cooler venues (A). Similarly, males with more slender body plans also ran relatively faster at hotter race venues compared to stockier male athletes (B). Results were not significant for female athletes in either case (C and D).

controversy stems in part from the paucity of studies designed to investigate causality underlying the relationships between body size, body shape, and temperature. The predominant hypothesis explaining both rules is that cooler temperatures should favor a reduction in an organism's surface area to volume ratio. For broadly distributed taxa this suggests that body shapes should be compact and appendages short in cold environments, but long and lean in hotter environments. I have shown that these predictions can also explain variation in endurance running performance for male, but not female, athletes.

Although these ecological rules were originally formulated to predict shifts in size and shape within species across broad geographic distributions [6], the results of this study (at least those pertaining to males) suggest that causal mechanisms underlying changes in body plan that are related to thermoregulation may act similarly across multiple spatial scales. Evolutionary processes transcend scale, hence the adage that 'macroevolution is just microevolution writ large' [23]. Although I do not envision athletic running performance as having a direct link to evolutionary fitness (but see [24]), I nevertheless propose a role for natural selection acting on morphology in populations of early humans via links between endurance running

**Table 2. Repeated-measures ANOVA tables for females (top) and males (bottom).** Variation in running performance consistent with Bergmann's rule (PC1 × temperature) and Allen's rule (PC2 × temperature) was significant for males but not females (grey-filled cells). Statistically significant effects are highlighted in bold.

| Females | swim | | bike | | Run | | total | |
|---|---|---|---|---|---|---|---|---|
| Variable | F-value | P-value | F-value | P-value | F-value | P-value | F-value | P-value |
| PC1 | 0.17 | 0.68 | 3.21 | 0.08 | 2.10 | 0.15 | **4.26** | **0.04** |
| temp | **5.24** | **0.03** | **4.12** | **0.05** | 1.58 | 0.21 | 0.16 | 0.69 |
| PC1×temp | 1.35 | 0.25 | 0.33 | 0.57 | 1.89 | 0.28 | 1.11 | 0.29 |
| PC2 | 0.01 | 0.99 | 0.14 | 0.71 | 0.57 | 0.45 | 0.19 | 0.66 |
| temp | **5.67** | **0.02** | **3.85** | **0.05** | **2.04** | **0.16** | 0.06 | 0.80 |
| PC2×temp | 0.01 | 0.98 | 0.58 | 0.45 | 0.29 | 0.59 | 0.05 | 0.82 |
| Males | swim | | bike | | Run | | total | |
| Variable | F-value | P-value | F-value | P-value | F-value | P-value | F-value | P-value |
| PC1 | 1.68 | 0.20 | 0.03 | 0.87 | 0.08 | 0.78 | 0.16 | 0.69 |
| temp | **10.97** | **0.002** | **27.3** | **<0.0001** | **4.76** | **0.03** | **15.05** | **0.0002** |
| PC1×temp | 1.78 | 0.19 | 2.29 | 0.13 | **9.65** | **0.003** | 7.38 | 0.01 |
| PC2 | 0.65 | 0.44 | 0.54 | 0.46 | **3.78** | **0.05** | 3.19 | 0.08 |
| temp | **14.98** | **0.0002** | **26.81** | **<0.0001** | **4.20** | **0.04** | **13.92** | **0.0003** |
| PC2×temp | 0.03 | 0.87 | 1.39 | 0.24 | **4.72** | **0.03** | 3.07 | 0.08 |

performance and fitness [25]. Changes in climate that accompanied human dispersal may have favored alternative limb morphologies and body shapes through their thermoregulatory impacts on endurance running. The mark of natural selection would be left in the form of human conformation to Bergmann's and Allen's rules, a pattern that has ample support from previous studies [4, 10, 12].

It is not surprising that performance variation was tied to morphology and temperature in the running event but not for cycling or swimming. Temperature is unlikely to affect performance in the swimming portion of Ironman in the same way that it affects running. Depending on water temperature, athletes may wear wetsuits during the swim, thereby mitigating the impact of water temperature on performance. Indeed, wetsuits provide increased insulation that mitigates the effects of temperature on performance. Likewise, increased air-flow at high cycling speeds should enhance evaporative cooling and at least partially reduce the influence of variation in thermoregulatory efficiency on performance [26]. Finally, while debate continues regarding whether endurance running played an important causal role in the evolution of human morphology, there is no doubt that swimming and cycling did not.

There are at least three possible explanations for the result that performance in this study varied with morphology and temperature in males but not females. The first is that sexual dimorphism in humans, in which males are on average larger than females, results in greater variance in morphological traits subject to selection. The tendency for trait distributions with larger mean values to have correspondingly larger variances could generate stronger correlations between performance, morphology and temperature in males than females. However, coefficients of variation in morphological traits were not consistently larger in males than females (Table 1), and I conclude that this is an unlikely explanation for the differences in our study.

Second, selection favoring improved running performance in early humans may have been largely limited to males. Early studies of bipedal human locomotion suggested that selection for improved running performance in males may have been antagonized by selection for improved obstetric outcomes in females (i.e., selection favored longer limb lengths in males but wider pelvic girdles in females) [27]. More recent studies discount the "obstetric dilemma"

[28], but suggest nevertheless that temperature affects male and female running performance differently. The higher percentage of subcutaneous fat in females compared to males, as well as the difference in density and distribution of sweat-glands compared to males may reduce the influence of body surface area for thermoregulation in females [29], suggesting that causal mechanisms underlying Bergmann's and Allen's rules act only on males. Indeed, longer/leaner females performed better than shorter/stockier females overall (Fig 2C and 2D) which may indicate a lower limit to the performance advantage afforded by reductions in surface area to volume ratios, at least over the range of temperatures measured in this study.

Finally, a third explanation is that Bergmann's and Allen's rule apply to both sexes, but patterns in the present study are only detected in males for reasons akin to an intraspecific version of Resch's rule [30]. Rensch's rule states that increases in body size are accompanied by allometric increases in the magnitude of sexual dimorphism [31]. In a meta-analysis of 98 published studies spanning vertebrates and invertebrates, Blanckenhorn et al. [32] showed that the majority (~2/3) of taxa that conformed to Bergmann's rule tended to show steeper latitudinal clines in males than females. The authors propose that allometric divergence in Bergmann clines was due to sexually antagonistic natural and/or sexual selection associated with environmental variation (i.e., temperature, food availability). Both human and non-human primates may exhibit patterns of sexual dimorphism consistent with Rensch's rule [33, 34] but see [35]. In addition, although some evidence indicates that human mating systems are causally related to temperature and food availability Alexander et al., (1979) [36] this idea also remains controversial [37, 38]. Although these hypotheses warrant further interrogation, results presented here are consistent with the idea that Rensch's, Bergmann's and Allen's rules could have interacted to create geographical variation in human morphology related to running performance.

## Acknowledgments

I thank C. Nadel, M. Gamble, R. Chandarana, M. Ayres, and M. McPeek for helpful discussions that improved this manuscript. T. Hilbun helped collate photographs.

## Author Contributions

**Conceptualization:** Ryan Calsbeek.

**Data curation:** Ryan Calsbeek.

**Formal analysis:** Ryan Calsbeek.

**Methodology:** Ryan Calsbeek.

**Writing – original draft:** Ryan Calsbeek.

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
