## [Decision Letter · Decision Letter 0]

16 Jan 2023

PONE-D-22-25621Ecological rules for global species distribution also predict performance variation in Ironman triathletes.PLOS ONE

Dear Dr. Calsbeek,

Thank you for submitting your manuscript to PLOS ONE. After careful consideration, we feel that it has merit but does not fully meet PLOS ONE’s publication criteria as it currently stands. Therefore, we invite you to submit a revised version of the manuscript that addresses the points raised during the review process.

We look forward to receiving your revised manuscript.

Kind regards,

Daniel de Paiva Silva, Ph.D.

Academic Editor

PLOS ONE

and https://journals.plos.org/plosone/s/file?id=ba62/PLOSOne_formatting_sample_title_authors_affiliations.pdf.

Additional Editor Comments:

Dear Dr. Calsbeek,

After this first review round both reviewers had a pleasant reading of the manuscript. They both think that your study is an interesting contribution to the field and an elegant form to apply macroecological theories to humans. Therefore, congratulations!

Considering it all, your study received a minor review. Please resubmit by March 16th, 2023, along with a rebuttal letter informing the reviewers and me about the improvements applied to the new version of your text.

Sincerely,

Daniel Silva, PHD

Reviewers' comments:

Reviewer's Responses to Questions

**Comments to the Author**

1. Is the manuscript technically sound, and do the data support the conclusions?

Reviewer #1: Yes

Reviewer #2: Yes

2. Has the statistical analysis been performed appropriately and rigorously? 

Reviewer #1: Yes

Reviewer #2: Yes

3. Have the authors made all data underlying the findings in their manuscript fully available?

Reviewer #1: Yes

Reviewer #2: Yes

4. Is the manuscript presented in an intelligible fashion and written in standard English?

Reviewer #1: Yes

Reviewer #2: Yes

5. Review Comments to the Author

Reviewer #1: I love this paper. It's a brilliant way to use Bergmann's rule, and the science is solid as far as I can tell.

The authors stated their raw data will be uploaded to Dryad. A line of text pointing to this deposited data, such as with a DOI or URL, is needed.

I found one typo, referring to "cursors." Should that be "precursors?"

Reviewer #2: In general, the work is well-written and adequately tests all proposed ideas. Also, I think it will be fascinating to show the effect of Bergman and Allen's rules on Homo sapiens. I have just a few suggestions. First, it would be interesting to relate the shape and size variables of the body parts with the athletes' running endurance using Structure Equation Modeling (SEM) between the effects of each rule on running performance. It could be added as supplementary material showing that there is also a significant direct/indirect effect of predictor variables (PC1, PC2 and delta temperature) and running performance. Second, wouldn't the performance pattern found in male triathletes be related to the evolution of human hunting? For example, humans varied in the types of hunting (i.e., big game hunters coming from the Clovis or El Jobo culture, and hunters and gatherers). This could be related to this morphological variation. I will not impose this path of discussion on the author, but it would be interesting to test this work a posteriori. Instead, I related the possible causes reported in the discussion with the evolution of hunting to find out which mechanisms may have directly/indirectly affected performance in the race.

I will leave specific suggestions below:

In the sentence "I included only athletes that had competed in at least two Ironman distance triathlons", why did you only include athletes that completed two distance triathlons? This can best be detailed briefly.

In the sentence "and for whom I could find high resolution images from the cycling portion of the event taken from an orthogonal angle to the direction of travel.", you could present the sampled figures as supplementary material and cite that supplementary material here.

In the sentence "Missing values for height were estimated from the from the regression of known heights on total leg length (r2=0.43)." I needed to understand how each height was obtained. How was this regression done? What are the variables being related? Also, "from the" is repeated. Please remove one of them.

In the sentence "This difference was statistically significant (PC1 X ΔT X sex, P<0.01)." it was confusing to say "statistically significant". I think the author should explain the contextual interpretation made of the statistic. It is redundant to say that the result was significant because p<0.01 already says so. This is repeated throughout the rest of the results.

6. PLOS authors have the option to publish the peer review history of their article (what does this mean?). If published, this will include your full peer review and any attached files.

Reviewer #1: **Yes: **Matan Shelomi

Reviewer #2: No

---

## [Author Response · Author response to Decision Letter 0]

30 Jan 2023

Reviewer #1: I love this paper. It's a brilliant way to use Bergmann's rule, and the science is solid as far as I can tell.

The authors stated their raw data will be uploaded to Dryad. A line of text pointing to this deposited data, such as with a DOI or URL, is needed.

I found one typo, referring to "cursors." Should that be "precursors?"

-Thank you so much for your kind words. I have added the Dryad DOI above. 

-“cursors” referred to cursorial mammals but for clarity I have reworded the sentence to read “runners”. 

Reviewer #2: In general, the work is well-written and adequately tests all proposed ideas. Also, I think it will be fascinating to show the effect of Bergman and Allen's rules on Homo sapiens. I have just a few suggestions. First, it would be interesting to relate the shape and size variables of the body parts with the athletes' running endurance using Structure Equation Modeling (SEM) between the effects of each rule on running performance. It could be added as supplementary material showing that there is also a significant direct/indirect effect of predictor variables (PC1, PC2 and delta temperature) and running performance. Second, wouldn't the performance pattern found in male triathletes be related to the evolution of human hunting? For example, humans varied in the types of hunting (i.e., big game hunters coming from the Clovis or El Jobo culture, and hunters and gatherers). This could be related to this morphological variation. I will not impose this path of discussion on the author, but it would be interesting to test this work a posteriori. Instead, I related the possible causes reported in the discussion with the evolution of hunting to find out which mechanisms may have directly/indirectly affected performance in the race.

-Yes I agree that the selection for running performance is most likely tied to human male involvement in persistence hunting. This is the topic of the cited Nature paper and I have clarified that point in the revised ms. 

I will leave specific suggestions below:

In the sentence "I included only athletes that had competed in at least two Ironman distance triathlons", why did you only include athletes that completed two distance triathlons? This can best be detailed briefly.

-Thank you, I have added a clarifying remark that this was necessary to make comparisons between events at two different temperatures. 

In the sentence "and for whom I could find high resolution images from the cycling portion of the event taken from an orthogonal angle to the direction of travel.", you could present the sampled figures as supplementary material and cite that supplementary material here.

-Two representative images have been added as a supplementary Figure 1 and referenced in the revision. Thank you for the suggestion. 

In the sentence "Missing values for height were estimated from the from the regression of known heights on total leg length (r2=0.43)." I needed to understand how each height was obtained. How was this regression done? What are the variables being related? Also, "from the" is repeated. Please remove one of them.

-I used the regression of published body heights on measured leg length to obtain a regression equation that was then used to calculate “predicted height”. I have clarified this point and removed the extra “from the”.

In the sentence "This difference was statistically significant (PC1 X ΔT X sex, P<0.01)." it was confusing to say "statistically significant". I think the author should explain the contextual interpretation made of the statistic. It is redundant to say that the result was significant because p<0.01 already says so. This is repeated throughout the rest of the results.

Here and elsewhere I have edited these statements. For example, the example above now reads “Together, there was an overall difference in how body size interacted with temperature to affect performance between the sexes”.

Thank you again to both referees and the editor for your help on this ms.

---

## [Decision Letter · Decision Letter 1]

6 Mar 2023

Ecological rules for global species distribution also predict performance variation in Ironman triathletes.

PONE-D-22-25621R1

Dear Dr. Calsbeek,

We’re pleased to inform you that your manuscript has been judged scientifically suitable for publication and will be formally accepted for publication once it meets all outstanding technical requirements.

Kind regards,

Daniel de Paiva Silva, Ph.D.

Academic Editor

PLOS ONE

Additional Editor Comments (optional):

Dear Dr. Calsbeek,

I am pleasured to inform you and your co-authors that your manuscript has been formally accepted for publication in PLoS One. Congratulations on your efforts to improve the text.

Sincerely,

Daniel Silva

Reviewers' comments:

Reviewer's Responses to Questions

**Comments to the Author**

1. If the authors have adequately addressed your comments raised in a previous round of review and you feel that this manuscript is now acceptable for publication, you may indicate that here to bypass the “Comments to the Author” section, enter your conflict of interest statement in the “Confidential to Editor” section, and submit your "Accept" recommendation.

Reviewer #1: All comments have been addressed

Reviewer #2: All comments have been addressed

2. Is the manuscript technically sound, and do the data support the conclusions?

Reviewer #1: Yes

Reviewer #2: Yes

3. Has the statistical analysis been performed appropriately and rigorously? 

Reviewer #1: Yes

Reviewer #2: Yes

4. Have the authors made all data underlying the findings in their manuscript fully available?

Reviewer #1: Yes

Reviewer #2: Yes

5. Is the manuscript presented in an intelligible fashion and written in standard English?

Reviewer #1: Yes

Reviewer #2: Yes

6. Review Comments to the Author

Reviewer #1: I am satisfied with the corrections. No further edits needed.

Reviewer #2: In general, you have solved all comments I made. Then, I am excited to read this paper after its publication. Congratulations, for me it is accepted!

7. PLOS authors have the option to publish the peer review history of their article (what does this mean?). If published, this will include your full peer review and any attached files.

Reviewer #1: No

Reviewer #2: No

---

## [Editor Report · Acceptance letter]

10 Mar 2023

PONE-D-22-25621R1 

Ecological rules for global species distribution also predict performance variation in Ironman triathletes. 

Dear Dr. Calsbeek:

I'm pleased to inform you that your manuscript has been deemed suitable for publication in PLOS ONE. Congratulations! Your manuscript is now with our production department. 

Kind regards, 

on behalf of

Dr. Daniel de Paiva Silva 

Academic Editor

PLOS ONE